# IL-Functionalized Mg_3_Al-LDH as New Efficient Adsorbent for Pd Recovery from Aqueous Solutions

**DOI:** 10.3390/ijms23169107

**Published:** 2022-08-14

**Authors:** Laura Cocheci, Lavinia Lupa, Nick Samuel Tolea, Radu Lazău, Rodica Pode

**Affiliations:** Faculty of Industrial Chemistry and Environmental Engineering, Politehnica University Timisoara, Blv. Vasile Parvan No. 6, 300223 Timisoara, Romania

**Keywords:** adsorption, double layered hydroxides, ionic liquids, palladium

## Abstract

Palladium is a noble metal of the platinum group metals (PGMs) with a high value and major industrial applications. Due to the scarce palladium resources, researchers’ attention is currently focused on Pd ions recovery from secondary sources. Regarding the recovery process from aqueous solutions, many methods were studied, amongst which adsorption process gained a special attention due to its clear advantages. Moreover, the efficiency and the selectivity of an adsorbent material can be further improved by functionalization of various solid supports. In this context, the present work aims at the synthesis and characterization of Mg_3_Al-LDH and its functionalization with ionic liquid (IL) (Methyltrialkylammonium chloride) to obtain adsorbent materials with high efficiency in Pd removal from aqueous solutions. The maximum adsorption capacity developed by Mg_3_Al-LDH is 142.9 mg Pd., and depending on the functionalization method used (sonication and co-synthesis, respectively) the maximum adsorption capacity increases considerably, q_max_-Mg_3_Al IL-US = 227.3 mg/g and q_max_-Mg_3_Al IL-COS = 277.8 mg/g.

## 1. Introduction

Palladium (Pd) is part of the platinum group metals (PGMs), which were identified by the European Commission as part of the 27 “critical” raw materials with economic importance [1]. Its chemical stability and good conducting properties represent the most important reasons why Pd is used in so many important fields, such as: electronic and telecommunication industries, catalytic applications, medicine, etc. Due to its intensive use in the last years, it could be found in higher concentrations in secondary sources compared with the concentrations from natural ores [2]. Thus, special attention was paid to reclaiming Pd from secondary sources, especially from wastewaters. One of the most efficient methods of Pd recovery from aqueous solutions is adsorption, due to its numerous advantages: high performance, ease of operation, low-cost, wide operability [3,4,5]. Wojnicki et al. recovered palladium (II) chloride complex ion by adsorption onto charcoal (maximum adsorption capacity q_m_ = 42.4 mg/g at 323 K) [6], Zhang et al. studied nanosilica modified with imidazolium, groups as adsorbent material for Pd recovery from aqueous solutions (q_m_ = 69.6 mg/g) [2], Chitosan and modified chitosan were also used as adsorbent materials [7,8]. The list of examples can go on, but the interest in finding the most effective adsorbent for recovering Pd ions from aqueous solutions is still relevant. The challenge in the adsorption processes is to use an adsorbent material with the following key features: high adsorption capacity, high output/efficiency, easy to synthesize, stable and reusable. All these characteristics are associated with layered double hydroxide materials. Layered double hydroxide materials (LDH) are lamellarstructure solids, also known as anionic clays, which can be synthetically prepared in a great number of possible composition and metal anions combinations. Unique and ordered structure, flexible tunability, high chemical and thermal stability, ability to intercalate different type of anions, are just a few properties, which make LDH superior to other adsorbent materials [9,10].

In the last decade, to enhance the adsorption properties of LDH materials, the researchers focused on their structure functionalization/modification with different pendant or functional groups. Tang et al. published a comprehensive review regarding the functionalization of LDH materials, through (i) intercalation of different molecules, (ii) surface modification, (iii) LDH loading on different substrates by in situ growth, which were used for the adsorption of various metal ions (Ag^+^, Pb^+^, Cu^2+^, Cd^2+^, Co^2+^, Zn^2+^, Ni^2+^) from aqueous solutions [11]. Another review about functionalized LDH used for anionic and cationic removal from aqueous solutions was published by Ishak et al. [12]. In conclusion, functionalized LDHs have better selective absorption capacity, stability, and recyclability. Ionic liquids are salts, in the liquid state at room temperature, which contain both an anion and a cation. Due to their useful physico-chemical properties, such as high thermal and chemical stability, low volatility, and very high ability to dissolve a wide range of compounds, they gained a particular attention for application as functional element for the structural modification of different materials. It was observed that by using ionic liquids, functionalized materials were obtained with superior adsorbent behavior in many applications: CO_2_ separation [13,14], dyes removal [15,16], metal ions adsorption [17,18]. ILs were also used for LDH functionalization/modification to obtain materials with superior catalytic performance [19,20,21], or improved anionic exchange membranes [22]. From our knowledge, there are no published data to discuss the functionalization of Mg_3_Al-LDHs with ionic liquids to be used as adsorbent materials for the removal of metal ions from aqueous solutions. In our recently published paper [23], we studied the sequential use of ionic liquid-functionalized Zn-Al layered double hydroxide as adsorbent and photocatalyst, which presented very good performance in both of the studied water treatment techniques. However, in a review published recently by Tang et al., the important role of HO^−^ group and valence state of [Mg(OH)_6_] octahedron in the synthesis of functionalized layered double hydroxides using various method (intercalation, surface loading or modification) is underlined [11].Therefore, the purpose of this paper was the obtaining and characterisation of a new efficient adsorbent material for palladium recover from aqueous solutions. The new adsorbent material consists from Mg_3_Al-LDH which was functionalized with Methyltrialkylammonium chloride ionic liquid. The functionalization of Mg_3_Al-LDH with the studied ionic liquid was carried by two methods: ultrasonic method and co-synthesis. The used ionic liquid was Methyltrialkylammonium chloride, based on the reports of its effect in improved adsorption properties of other impregnated solid supports, such as chitosan, Zn_3_Al, or alginate, used for removal of lead ions, dyes or phenol from aqueous solutions [24,25,26,27]. The paper aims the determination of the efficiency of the new synthesized adsorbents materials, the influence of the ionic liquids and the influence of the synthesis method upon the adsorption capacity developed by the synthesized materials in the recovery process of palladium ions from aqueous solutions.

## 2. Results and Discussion

### 2.1. Adsorbent Materials Characterization

The X-ray diffraction patterns (Figure 1) were recorded to confirm the phase composition of the synthesized Mg_3_Al layered double hydroxides and to observe the IL functionalization effect upon samples structure and crystallinity. The XRD pattern of the Mg_3_Al sample showed characteristic peaks of hydrotalcite (JCPDS-41-1428) formed from symmetric, intensive and sharp peaks at low 2θ angle (11.2° and 22.6° and 34.4°), respectively from asymmetric and broad peaks at high 2θ angle (60.3° and 61.5°), suggesting a good sample crystallization. These results are similar with other reported data in literature [28,29,30].

The calculated lattice parameter, c, offers information about the interlayer distance of the LDH. The values presented in Table 1 clearly show the CO_3_^2−^ intercalation in the Mg_3_Al IL-US (c = 23.52 Å), as compared to Mg_3_Al sample (c = 23.58 Å) [31].

The IL functionalization of Mg_3_Al by ultrasound method does not generate a disorder of the LDH structure. In case of Mg_3_Al IL-COS could be observed a shift of the main peaks at lower 2θ values (10.7°, 21.9° and 34.3°). The c parameter for Mg_3_Al IL-COS is higher, c = 24.39 Å suggesting that through functionalization of Mg_3_Al by co-synthesis the intercalation of IL between the LDH interlayer took place, which also led to a slight decrease in the crystallite size (Table 1). For all three samples the typical doublet at approximately 60 degrees 2θ could be observed, and the calculated values of a parameters for (110) planes are around 3.07 Å, which is in good agreement with other data presented in the literature for similar materials [32,33]. The fact that in case of Mg_3_Al functionalization through ultrasound method the order of the LDH structure is not disturbed, the studied IL is found on the surface of the LDH, while in case of the co-synthesis method, appears a structure disturbance, the IL being between the LDH layers was evidenced by the morphological analysis of the sample published in our previous work [34].

The thermal analysis (Figure 2) shows the behavior of the samples during the thermal treatment up to 1000 °C. For the Mg_3_Al sample, the main thermal decomposition processes (dehydration, dehydroxylation and decarbonization) occur in the temperature intervals: 20–300, 300–700 and above 700 °C. Between 20 and 300 °C, the DSC diagram of Mg_3_Al sample reveals an unresolved endothermic shoulder at approximately 85 °C, due to the loss of adsorbed water and an endothermic peak at 188 °C which is due to the loss of interlayer water molecules (dehydration) and partial dehydroxylation (some water molecules formed in the recombination of hydroxyl groups from the brucite-like layer could be removed at this temperature) [35]. The mass loss in this temperature interval is 17.6%. Between 300 and 700 °C, a consistent mass loss takes place (28.0%) accompanied by an endothermic peak at 395 °C caused by the total dehydroxylation and partial decarbonization of the LDH. A broad endothermic peak and another endothermic peak with maxima at about 740 °C and at 905 °C, respectively, are due to the total decarbonization of the material and the formation of mixed metal oxides. The total mass loss due to the decomposition of Mg_3_Al sample is 45.7%.

By comparison, for the Mg_3_Al IL-US sample, the DSC curve reveals the unresolved endothermic shoulder due to the loss of adsorbed water at approximately 70 °C, another endothermic shoulder at 166 °C and an endothermic peak at 206 °C. The presence of two peaks due to the loss of interlayer water molecules and partial dehydroxylation, instead of one, and the displacement (shifting) of the maximum temperature to lower values for the shoulders and at higher value for the peak (as compared to sample Mg_3_Al) could be a consequence of the interaction of the organic IL with the physically adsorbed water molecules and those placed in the interlayer space of LDH. The mass loss for this temperature interval is 19.1%, greater than the mass loss of Mg_3_Al sample, suggesting that IL interacts also with some hydroxyl anions from the LDH layer, forcing the dehydroxylation process to occur at lower temperature. The sharp exothermic peak with maximum at 251 °C and the unresolved exothermic shoulder at about 300 °C appear due to the combustion of organic chain of ionic liquid. This exothermic peak overlapped to the endothermic peak with maximum at 402 °C, which is due to the dehydroxylation and decarbonization of the LDH. The mass loss for this temperature interval is 30.2%. The broad endothermic peak at approximately 630 °C and the small shoulder at about 900 °C are due to the formation of mixed metal oxides. The total mass loss for Mg_3_Al IL-US material is 49.3%.

The appearance of thermoanalytic curves of Mg_3_Al IL-COS sample is slightly different from those of the other two samples. Due to the fact that the ionic liquid is placed in the interlayer space, it probably replaced water molecules in this space, and interacted with carbonate anions placed in the interlayer space and with the hydroxyl anions on the brucite-like layer. Thus, in the first temperature range (20–200 °C), the DSC curve shows a single endothermic peak with the maximum at 129 °C, instead of two or three endothermic peaks as the other samples had. Additionally, the mass loss for this temperature range is the lowest (12.5%). The exothermic peak due to the combustion of the ionic liquid shifted to a higher temperature than in the case of the Mg_3_Al IL-US sample (256 °C vs. 251 °C), suggesting that a higher energy was needed to break some interactions between IL and LDH. Also, the endothermic peak due to the dehydroxyation and decarbonization of the material is broader, its maximum being observed between 375 and 396 °C, it is followed by an endothermic peak at 480 °C, and the mass loss in this second temperature range (200–700 °C) is the highest, 35.3%. At temperatures higher than 700 °C, an endothermic shoulder appears due to the formation of mixed, crystalline metal oxides. The total mass loss of this material is 47.9%.

FTIR spectra of the synthesized samples (Figure 3) highlights the presence of carbonate anions and water from the interlayer space, as well as the type of bonds formed in the studied materials. Information can also be obtained on how ionic liquid interacts with layered double hydroxide, depending on the method of materials synthesis (by ultrasonication or co-synthesis).

The spectrum of ionic liquid Methyltrialkylammonium chloride (Figure 3a) exhibits two absorption bands situated at 2920 cm^−1^ and 2852 cm^−1^ assigned to antisymmetric and symmetric stretching vibration of -CH_2_- and -CH_3_ groups, respectively. Usually, the bending vibration of C-H bond of the quaternary ammonium salt with longalkyl chains give two absorption bands near 1460 cm^−1^ specific for C–H bending of both -CH_2_- and -CH_3_ groups. The band with the maximum at 1377 cm^−1^ corresponds to the -CH_2_- trans-plane bending vibration [36].

The Mg_3_Al FTIR spectrum (Figure 3b) contains the absorption bands that are expected from a layered double hydroxide sample with carbonate as interlayer anion. The broad band at about 3500 cm^−1^ is the result of several bands that are partially overlapping in this region: one band at 3555 cm^−1^ that could be due to the hydroxyl coordinated by both Mg an Al, another band at 3470 cm^−1^ that could be assigned to the Al-OH bond and a shoulder at 2976 cm^−1^ assigned to CO_3_^2−^-H_2_O bridging mode of carbonate and water in the interlayer region [37]. The absorption band situated at 1641 cm^−1^ is attributable to the bending mode of water molecules from the interlayer region. The band at 1373 cm^−1^ and the shoulder at 1496 cm^−1^ could be attributed to the ν_3_ mode (antisymmetric stretching) of interlayer carbonate anion, the splitting of the band being a sign of the low symmetry of the anion in the interlayer space, probably due to hydrogen bonding with hydroxyl groups or with water molecules [38]. The two bands at 854 cm^−1^ and 658 cm^−1^ are characteristic to the ν_2_ and ν_4_ vibration modes of carbonate group. The absorption band situated at 412 cm^−1^ is attributable to the vibration mode of the hydrotalcite octahedral network [39].

The spectrum of Mg_3_Al IL-US sample (Figure 3c) displays all the bands observed in the Mg_3_Al sample. The only noticeable differences between the FTIR spectrum of Mg_3_Al and Mg_3_Al IL-US sample are: the presence of the absorption bands assigned to -CH_2_- and -CH_3_ groups in IL (observed at 2926 cm^−1^ and 2856 cm^−1^), that overlap with the band assigned to the CO_3_^2−^-H_2_O (bridging mode of carbonate and water in the interlayer region), and the shift of the band attributed to the bending mode of water molecules from 1641 cm^−1^ (observed in the spectrum of Mg_3_Al sample) to 1628 cm^−1^. This shift could be a consequence of the weakening of the hydrogen bonding between water and hydroxide layer due to the hydrophobicity of ionic liquid. This weakening of the bonds between water and hydroxide layer is in accordance to the findings on thermoanalysis, where the DSC curve of Mg_3_Al IL-US sample (Figure 2) suggested that the loss of interlayer water occured at lower temperature. Due to the different synthesis mode, the Mg_3_Al IL-COS sample, shows several changes in the FTIR spectrum (Figure 3d), as a consequence of the presence of ionic liquid in the interlayer space. The band at about 3500 cm^−1^ is broader, probably due to the interaction between IL and the hydroxide layer, which causes the Mg-OH and Al-OH bonds to change their length. The doublet assigned to -CH_2_- and -CH_3_ groups in IL is found at 2932 cm^−1^ and 2864 cm^−1^. Due to this doublet, the band assigned to the carbonate-water bridging mode is not visible. Moreover, the band attributed to the bending mode of water molecules is found shifted to 1628 cm^−1^ and a new shoulder situated at 1759 cm^−1^ appears on the spectrum. The presence of this new shoulder can be explained by the presence of small amounts of coordinated water molecules to a cation [38]. This supposition is in accordance with the thermoanalytic studies, the DSC curve of Mg_3_Al IL-COS sample showing a single endothermic peak due to the water loss, a sign of structural changes brought by the hydrophobic ionic liquid. Also, the presence of IL in the interlayer region is demonstrated by the shift of the band assigned to ν_2_ vibration mode of carbonate group, from about 850 cm^−1^ (in Mg_3_Al and Mg_3_Al IL-US samples) to 824 cm^−1^, which could be explained by a change in the interaction between carbonate anion present in the interlayer space and the hydroxide layer. The endothermic shoulder at 480 °C found only on the DSC curve of the material (Figure 2), was attributed to the decarbonization process and could be a consequence of this change in the carbonate anion vibration mode.

The adsorption-desorption isotherms of the synthetized materials are presented in Figure 4. The BET surface areas calculated for this samples are presented in Table 1 and decrease as follows: Mg_3_Al > Mg_3_Al IL-US > Mg_3_Al IL-COS. Regardless of the obtaining method, all isotherms belong to the Type IV according to IUPAC classification and belong to the H3 type of hysteresis loop. This kind of isotherm is associated with capillary condensation taking place in mesopores, does not exhibit any limiting adsorption at high p/p_0_ and is observed with aggregates of plate-like particles giving rise to slit-shaped pores. A difference is observed, however, between materials Mg_3_Al and Mg_3_Al IL-US on the one hand, and material Mg_3_Al IL-COS, on the other hand, in terms of N_2_ volume adsorbed, which could be correlated to the pore diameter distribution. For samples Mg_3_Al and Mg_3_Al IL-US, the mesopores, with dimensions between 10 and 20 nm have major contributions to the textural properties, while the distribution range of the pore diameter of Mg_3_Al IL-COS sample is wider, large mesopores and macropores (with dimensions between 30 and 70 nm) having the major contribution.

### 2.2. Kinetic Studies

The adsorption studies conducted at different stirring time led to the conclusions that the equilibrium between the Mg_3_Al with or without Methyltrialkylammonium chloride and Pd(II) ions from the aqueous solutions, regardless of the work temperature, was achieved quite fast, in 60 min. The dependence of the adsorption capacities developed by the studied adsorbent materials function of stirring time, at three different temperatures are presented in Figure 5. It is observed that the impregnated LDHs present a higher adsorption capacity than the raw LDH. In the case of non-functionalized Mg_3_Al, at the working temperature of 298 K, a degree of 75.6% of Pd (II) ions recovering from solution is achieved, while in the case of the other two materials the removal efficiency is over 95%. The removal efficiency increases with the temperature increasing. Using the sample obtained through co-synthesis, a 100% removal efficiency is obtained at 313 K, and in case of the sample obtained through ultrasonication a complete removal efficiency is obtained at 328 K.

The experimental results were modelled with pseudo-first order, pseudo-second order and intra particle diffusion establish the rate limiting step and the type of adsorption mechanism.

The pseudo-first order model is expressed in its linear form, according to the following equation [40,41,42].
(1)$$\ln\left(q_{e}-q_{t}\right)=\ln q_{t}-k_{1}\cdot t$$
where: q_e_ and q_t_ represent the amount of Pd(II) adsorbed onto 1 g of adsorbent materials at equilibrium and after the treatment time, t; k_1_ represent the rate constant specific for the adsorption process of Pd ions onto studied materials.

The rate constant and the equilibrium adsorption capacity were determined from the slope and intercept of ln(q_e_ − q_t_) versus stirring time (Figure 6).

If the process of Pd(II) adsorption onto the studied LDHs is controlled by the chemisorptions rate, then the experimental data are well modeled by the pseudo-second order kinetic model in its linear form:(2)$$\frac{t}{q_{t}}=\frac{1}{k_{2}\cdot q_{e}^{2}}+\frac{t}{q_{e}}$$
where: q_e_, q_t_ and t have the same significance as shown above, and k_2_ is the rate constant of the pseudo-second order kinetic model.

These parameters could be determined from the slope and intercept of the linear plot between t/q_t_ and stirring time (Figure 7).

The rate limiting step in the process of adsorption of Pd ions onto studied LDHs could be established from the intraparticle diffusion experimental data model:(3)$$q_{t}=k_{\mathrm{in}}\cdot t^{\frac{1}{2}}+C$$
where: k_in_ represent the rate constant of the intraparticle diffusion model and was determined from the slope of the linear representation of q_t_ versus t^1/2^ (Figure 8).

The calculated rate constants, together with the obtained correlation coefficient are presented in Table 2.

Data presented show that the kinetics of Pd(II) removal by adsorption on the studied material is described by a pseudo-second-order expression, because the correlation coefficient is very close to 1 and the theoretically predicted equilibrium sorption capacity is close to the value experimentally determined. This suggests that the rate-determining step may be chemical adsorption or chemisorption involving valence forces through sharing or exchange of electrons between adsorbent and adsorbate. For all the studied materials, the pseudo-second order constant rate, k_2_ increases with the temperature increasing, suggesting that the Pd adsorption onto the studied materials is favored by the temperature increasing. This chemisorption process is a complex one, which takes place in two steps, according to the intraparticle diffusion plot (Figure 8). It is obvious that the intraparticle diffusion is not the rate-limiting step for adsorption of Pd(II) onto the studied LDH. In the first 10 min the Pd(II) diffuses through the solution until it reaches the LDHs surface, then the rate limiting step is controlled by the adsorption of the Pd(II) inside the particles of raw or functionalized Mg_3_Al [43].

### 2.3. Thermodynamic Studies

To further elucidate if the Pd adsorption onto Mg_3_Al-LDH and the samples functionalized with the studied ionic liquid, correspond to a physisorption or a chemisorption process, the activation energy was determined according to the Arrhenius equation described by the following equation:(4)$$\mathrm{lnk}=\mathrm{lnA}-\frac{E_{a}}{\mathrm{RT}}$$

Activation energy in the adsorption process is the minimum energy needed of the adsorbate to interact with the functional groups on the adsorbent surfaces. If the interactions between the adsorbent and adsorbate involve week forces, an activation energy bellow 4.2 kJ/mol is obtained meaning that a physisorption process takes place. If the interactions between the adsorbent and adsorbate involves stronger forces, higher values of E_a_ are obtained suggesting that a chemisorption process takes place [44,45]. The activation energy E_a_ for the adsorption process of Pd ions onto the studied materials were determined from the plot of lnK_2_ versus 1/T (Figure 9) and the results are presented in Table 3. It could be observed that in case of Mg_3_Al an activation energy of 2.97 kJ/mol is obtained, suggesting that Pd adsorption onto LDH take please only as physisorption in the materials pores.

In case of the LDH functionalized Methyltrialkylammonium chloride for the sample obtained through co-synthesis, an E_a_ = 14.72 kJ/mol is obtained, and for the one obtained through ultrasonication an E_a_ = 10.68 kJ/mol, respectively. In these cases, it is worth mentioning that the amino group introduced in the LDH structure contribute to the Pd recover from aqueous solutions involving stronger forces, and suggesting that Pd adsorption takes place due to a chemisorption process.

To conclude, if the Pd adsorption onto the studied LDHs is a spontaneous process or not, the Gibb’s free energy (ΔG°) was determined, taking into consideration also the enthalpy (ΔH°) and the entropy (ΔS°) of the Pd adsorption process. The Gibb’s free energy (ΔG°) was determined from the classical Van’t Hoff equation [44,45].
(5)$$∆G^{0}=-{\mathrm{RTlnK}}_{d\;}$$
(6)$$∆G^{0}=∆H^{0}-T∆S^{0}$$

K_d_, the equilibrium constant, could be determined at different temperature by the following equation:(7)$$K_{d}=\frac{q_{e}}{C_{e}}$$

The thermodynamic parameters of ΔH° and ΔS° were calculated from the linear regression according to the following equation represented in Figure 10:(8)$${\mathrm{lnK}}_{d}=\frac{∆S^{0}}{R}-\frac{∆H^{0}}{\mathrm{RT}}$$

The obtained thermodynamic parameters are listed in Table 3.

From the data presented in Table 3, for all the studied LDHs the positive values of enthalpy indicate the endothermic nature of Pd(II) adsorption. The values obtained for Gibbs free energy are negative and increase in module with the temperature increasing suggesting that the Pd(II) adsorption onto the studied LDHs is a spontaneous processes. For the entropy, a positive value was obtained suggesting an increase in randomness at the solid/liquid interface during adsorption of Pd(II) ions onto the studied LDHs.

### 2.4. Equilibrium Studies

The maximum adsorption capacity of raw and functionalized Mg_3_Al developed for the recovery process of Pd(II) from aqueous solutions could be experimentally determined from the equilibrium isotherm presented in Figure 11. It can be observed that the adsorption capacity of the studied materials increase with the equilibrium concentration of Pd(II) ions from the aqueous solutions. The adsorption capacity developed by the functionalized Mg_3_Al is almost two times higher (in case of the co-synthesis—Mg_3_Al IL-COS) than those developed by the raw Mg_3_Al.

Three isotherm models, based on two parameters (Langmuir, Freundlich and Temkin) were applied to model the equilibrium data obtained experimentally.

The linearized Langmuir isotherm represented by the following equation suggests that the Pd(II) adsorption onto studied LDHs occurs as a monolayer, covering the adsorbent surface:(9)$$\frac{C_{e}}{q_{e}}=\frac{1}{q_{m}\cdot K_{L}}+\frac{C_{e}}{q_{m}}$$
where: q_m_ represents the maximum adsorption capacity developed by the studied LDHs and K_L_ represents the Langmuir constant. These two parameters can be determined from the linear plot of C_e_/q_e_ versus C_e_ (Figure 12).

The affinity between the raw Mg_3_Al or functionalized Mg_3_Al and the Pd(II) ions can be obtained by fitting the data to the linearized form of the Freundlich isotherm:(10)$$\ln\left(q_{e}\right)=\ln(k_{F})+\frac{1}{n}\ln(C_{e})$$
where: 1/n and the Freundlich constant k_F_ represent the Freundlich isotherms parameters and can be determined from the plot of ln(q_e_) versus ln(C_e_) presented in Figure 13.

If the surface of the adsorbent is heterogeneous, the equilibrium data will present a good fit for the linearized form of the Temkin isotherm model, which is given by the next equation, suggesting that the heat of adsorption decreases during the adsorption process:(11)$$q_{e}=\frac{R\cdot T}{b}\ln k_{T}+\frac{R\cdot T}{b}\ln(C_{e})$$
where: k_T_ is the equilibrium binding constant, and b is related to the heat of adsorption, which were determined from the linear plot between q_e_ and ln(C_e_) presented in Figure 14.

The equilibrium parameters together with the obtained correlation coefficient are presented in Table 4.

It could be observed that the Langmuir isotherm better describes the adsorption process of Pd(II) ions onto the studied material, as compared with other isotherm models due to the values of the obtained correlation coefficients, which are close to 1. Moreover, the maximum adsorption capacity of the studied material obtained from the Langmuir plot is very close to that obtained experimentally. The Langmuir model assumes that the surface of the adsorbent is homogenous and the sorption energy to be equivalent for each sorption site. Due to the fact that the Langmuir isotherm fits the experimental data better than the other isotherm types, it can be mentioned that the Pd(II) ions are adsorbed uniform onto the surface of the studied adsorbent, because of the homogenous distribution of active site onto the surface. In this case there is no migration of the palladium ions onto the surface of the studied adsorbent, suggesting a possible chemisorption process between the adsorbent and adsorbate.

### 2.5. Mechanism of Palladium Adsorption onto the Studied Materials

Correlating the results from the characterization section with the results obtained from kinetic, thermodynamic and equilibrium studies the following mechanism of palladium adsorption onto the studied materials could be proposed (Figure 15).

In case of Mg_3_Al, the Pd adsorption took place on the surface of the LDH, in pores or through electrostatic interactions between the Pd ions and the hidroxil ions presents on the surface of the LDH. This physical sorption of Pd recover from aqueous solutions, using Mg_3_Al as adsorbent material, is confirmed by the low value of activation energy, 2.97 kJ/mol, resulted from the thermodynamic study.

In case of the functionalized Mg_3_Al samples with Methyltrialkylammonium chloride obtained higher adsorption capacities due to the interaction of Pd ions, with the functional groups from the studied IL structure. As can be seen from Figure 15 could appear ion-part interaction between the Pd (which is found under palladium chloride) and the quaternary ammonium from the studied IL. These results are in agreement with other studies presented in literature [46,47,48,49]. It is also well known that the interaction between palladium ions and quaternary ammonium is highly used in the formation of stable complex which are used as catalyst in various chemical processes [50,51]. Therefore, the samples functionalyzed with Methyltrialkylammonium chloride developed higher adsorption capacities and higher values for activation energy. All the results from thermodynamic, kinetic and equilibrium studies underline the chemisorption process which describe the palladium recover from aqueous solutions onto Mg_3_Al IL-US, and Mg_3_Al IL-COS.

It could be observed that the synthesis method used for the adsorbent obtaining also influenced the adsorption capacity developed by the studied materials in the recovery process of palladium ions from aqueous solutions. The highest adsorption capacity is developed by the sample functionalized through co-synthesis. In this case, as the adsorption of palladium ions contribute as a synergic effect both the ionic liquid and also the Mg_3_Al–LDH. In this case the ionic liquid is found in the interlayer of the Mg_3_Al–LDH, and Pd ions are retained by the ion-pair and charge interaction with the IL, but also by physical sorption onto the surface of the LDH. In case of the sample obtained through ultrasound method, because the ionic liquid is found on the surface of the Mg_3_Al–LDH and not in the interlayer space, the Pd ions are retained by the interaction between the IL and it, but the adsorption process is decreased, because the loading of IL onto the surface of LDH decrease the available site for the Pd physical sorption.

### 2.6. Comparison of the Adsorption Capacity Developed by the Studied Materials with other Materials Reported in Literature

Comparing the maximum adsorption capacities developed by the studied adsorbents with the adsorption capacities of other materials, reported in literature, it could be concluded that the Mg_3_Al–LDH presents a higher efficiency, especially if it is functionalized with ionic liquid, in the removal process of Pd ions from aqueous solutions (Table 5).

The comparison of the adsorption capacities developed by the synthesized and characterized materials was made with other similar materials. It is observed that other researchers were also focused on the use of some functionalized materials and used ammonium as functional group. It was used the same ionic liquids Aliquat-336, but impregnated onto chitosan [48], or Zn_3_Al [23]. In these cases, we obtained an adsorption capacity of 187.61 mg/g and 100 mg/g, respectively, compared with the q_max_ = 277.3 mg/g developed by Mg_3_Al IL-COS. These results suggested that both the ionic liquid and also the support materials contribute as a synergic effect to the recovery of palladium ions from aqueous solutions. Another important result reported in the literature is the case of the use of tetraoctylammonium bromide impregnated onto graphene oxide. In this case, an adsorption capacity of 92.67 mg/g is obtained, suggesting the important role of the ammonium functional group in the recover process of Pd ions from aqueous solution due to the formation of the strong interactions between these two. Other reported materials present much more lower adsorption capacities, underlying one more time the importance of the properties of the used solid support and the efficiency demonstrated by the ammonium group.

## 3. Materials and Methods

### 3.1. Adsorbent Materials Obtaining and Characterization

The layered double hydroxide used in this study was synthetized by co-precipitation method using as precursors Mg(NO_3_)_2_ · 6H_2_O and Al(NO_3_)_3_ · 9H_2_O. The ratio between the cation in the LDH was calculated to be Mg:Al = 3:1. 200 mL of 1 M solution containing the nitrates salts in the desired metal ions ratio was prepared and added gradually, under vigorous stirring in a volume of 100 mL of 1M Na_2_CO_3_ solution. A 2M NaOH solution was used to maintain the suspension pH around 10.5 value. To obtain the Mg_3_Al-LDH, the suspension was subject to maturation for 24 h at 70 °C. The obtained precipitate was filtered, washed, crushed, sieved, and used in the further experiments. A part of the obtained Mg_3_Al was functionalized with the studied ionic liquid through ultrasound method, which was described in our previously published papers [23,53]. The obtained sample, symbolized Mg_3_Al IL-US, was prepared with 10% IL addition. For the second method of functionalization by co-synthesis, the studied ionic liquid was introduced in the stage of the LDH co-precipitation (Mg_3_Al IL-COS), by replacing the Na_2_CO_3_ solution with the IL previously dissolved in acetone [23]. The further steps of sample preparation were maintained. The obtained adsorbent materials were subjected to structural and morphological characterization using X-ray diffraction (XRD), N_2_ adsorption-desorption analysis, Fourier-transform infrared (FTIR) spectroscopy and by thermal analysis. The XR diffractograms were obtained by using a Rigaku Ultima IV X-ray diffractometer (40 kV, 40 mA) with Cu_K__α_ radiation. The N_2_ adsorption-desorption experiments were developed at 77 K by using a Micromeritics ASAP 2020 instrument, after degassing the samples for 48 h at 100 °C, under 8 μmHg vacuum. The specific surface area was calculated using the BET method and the pore size distribution was calculated using the BJH method based on the N_2_ desorption isotherm. The FTIR spectra of the samples were recorded using a Shimadzu IR Prestige-21 FTIR spectrophotometer in the range 400–4000 cm^−1^ and a nominal resolution of 4 cm^−1^, the samples being prepared as pellets by mixing with KBr. Thermal analysis was performed by using a Netzsch STA 449C equipment. Differential scanning calorimetry (DSC) and thermogravimetric (TG) curves were obtained by heating the materials in the temperature range 25–1000 °C, with a heating rate of 10 °C/min under air flow.

### 3.2. Pd Recover from Aqueous Solutions

The obtained Mg_3_Al–LDH with or without Methyltrialkylammonium chloride were used as adsorbent materials in the recovery process of Pd ions from aqueous solutions. The adsorption of palladium (II) ions was performed in batch mode. The efficiency of raw or impregnated Mg_3_Al-LDH was determined by studying the dependence of their developed adsorption capacity function of stirring time and initial concentration of Pd ions from aqueous solutions. From the literature survey it was concluded that the Pd ions are better adsorbed at pH values lower than 4 [53,54,55,56]. Therefore, in this paper all the adsorption studies were performed using Pd solutions with an initial pH around 3.6, the lowest possible pH to avoid the LDH dissolution. To determine the time when the equilibrium between the adsorbent and adsorbate is established, adsorption experiments were performed using different times (5–180 min) for stirring the reaction mixture, which consists of 25 mL of 50 mg/L Pd(II) solutions and 0.025 g of adsorbent materials. The kinetic studies were performed at three different temperatures (298 K, 313 K and 328 K). To determine the efficiency of the studied materials in the recovery process of Pd ions from aqueous solutions, their maximum adsorption capacity was established by performing equilibrium studies. In this way, the adsorption studies were conducted using Pd(II) solutions containing different initial concentrations (5–500 mg/L Pd(II)). The equilibrium studies were performed using the same initial pH of Pd solutions, the same solid:liquid ratio, S:L = 1:1, and a stirring time of 60 min. After stirring the mass reaction, the samples were filtered and in the resulted solutions the residual concentration of Pd(II) ions was determined through atomic absorption spectrophotometry, using an atomic absorption spectrophotometer Varian SpectrAA 280 FS.

The adsorption efficiency of the studied materials was expressed by the amount of Pd ions adsorbed on 1 g of adsorbent according to the following equation:(12)$$q_{e}=\frac{\left(C_{0}{\;-\;C}_{e}\right)V}{m}$$
where: q_e_—equilibrium capacity developed by adsorbent materials, mg Pd/g adsorbent material.
C_0_—initial concentration of Pd(II) in aqueous solutions, mg/L.C_e_—equilibrium concentration of Pd(II), mg/L.V—volume of the solution containing Pd(II) used in the adsorption process, L.m—mass of adsorbent material used in the adsorption process, g.

## 4. Conclusions

In the present work, Mg_3_Al-type layered double hydroxide was synthesized and functionalized by two methods (co-synthesis and ultrasonication), with ionic liquid (Methyltrialkylammonium chloride). X-ray diffraction, FTIR, BET and thermal analysis confirmed the formation of the Mg_3_Al-type layered double hydroxide.

By its functionalization by co-synthesis, IL is observed to be located between the LDH layers, and by the ultrasound method IL is found on the LDH surface.

The synthesized and characterized materials were used in the recovery process of Pd(II) ions from aqueous solutions. The adsorption capacity increases with the increasing contact time, increasing temperature and increasing equilibrium concentration of Pd(II). The presence of ionic liquid significantly improves the efficiency of Mg_3_Al in the recovery process of Pd(II) ions from aqueous solutions.

The adsorption kinetics were better described by the pseudo-second-order kinetic model compared to the pseudo-first-order kinetic model.

The experimental data showed a good fit with the Langmuir isotherm.

The adsorption capacity increases as follows: Mg_3_Al << Mg_3_Al IL-US < Mg_3_Al IL-COS.

By correlating the results obtained in the characterization process of adsorbent materials with the results obtained from kinetic, thermodynamic and equilibrium studies, we can conclude that in the case of the raw Mg_3_Al sample, Pd recovery occurs through a physisorption mechanism, as it is adsorbed in the pores of the material. In contrast, in the case of IL (Methyltrialkylammonium chloride) functionalized samples, Pd recovery is due to a chemosorption process, indicating that the functional groups of the ionic liquid confer a beneficial influence on the adsorbent material. This is very important for the future perspective of using and reclaiming of the exhausted adsorbent as photocatalyst in the degradation process of organic compounds from waters. Additionally, as a future experiment, the column adsorption test will be conducted in order to optimize the recover process of the Pd ions from aqueous solutions and in order to obtain a higher quantity of the photocatalysts. The use of exhausted adsorbent as a photocatalyst could close the life cycle of products through sustainable consumption and production by improving waste management with greater recycling and reuse to create benefits for the environment, the economy and society.

## Figures and Tables

**Figure 1 ijms-23-09107-f001:**
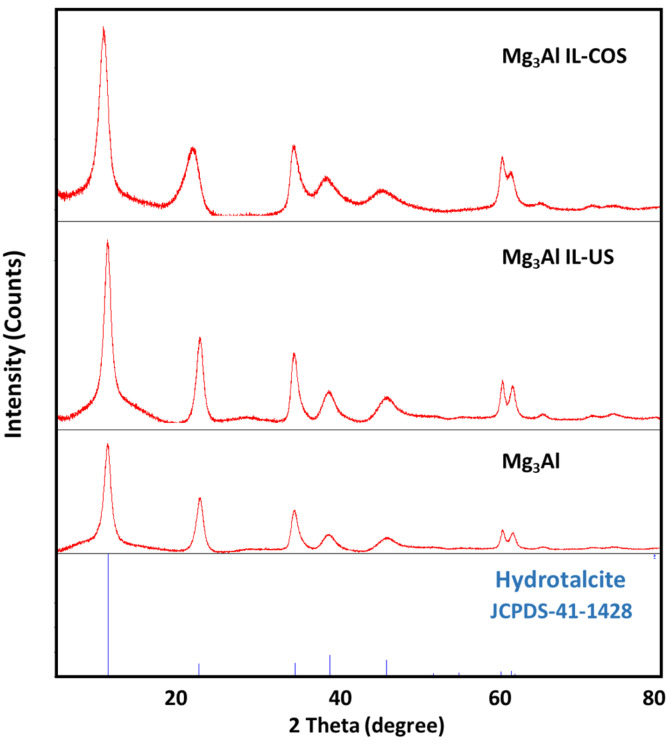
XRD patterns of the studied adsorbent materials.

**Figure 2 ijms-23-09107-f002:**
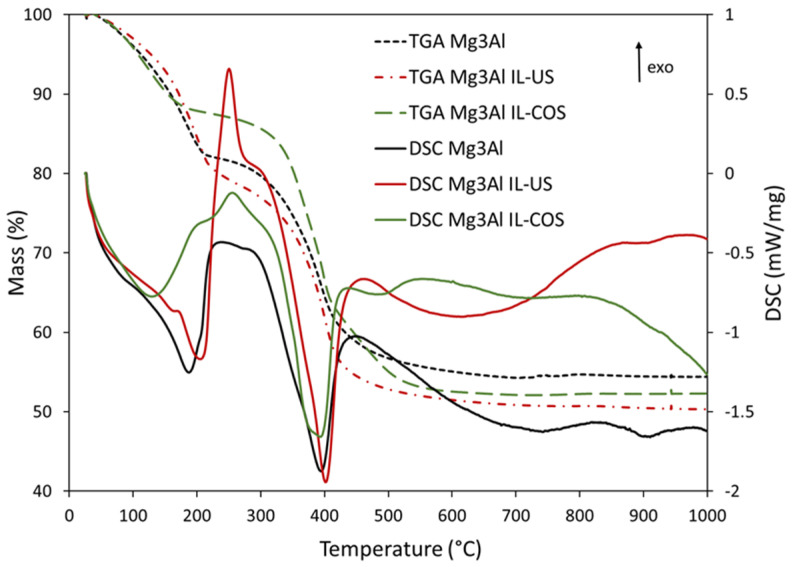
Thermoanalytic curves of the synthetized samples.

**Figure 3 ijms-23-09107-f003:**
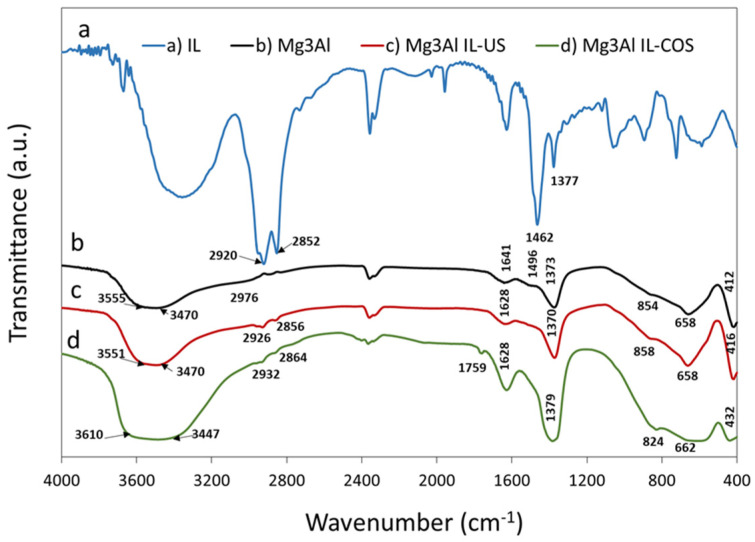
FTIR spectra of the synthesized samples.

**Figure 4 ijms-23-09107-f004:**
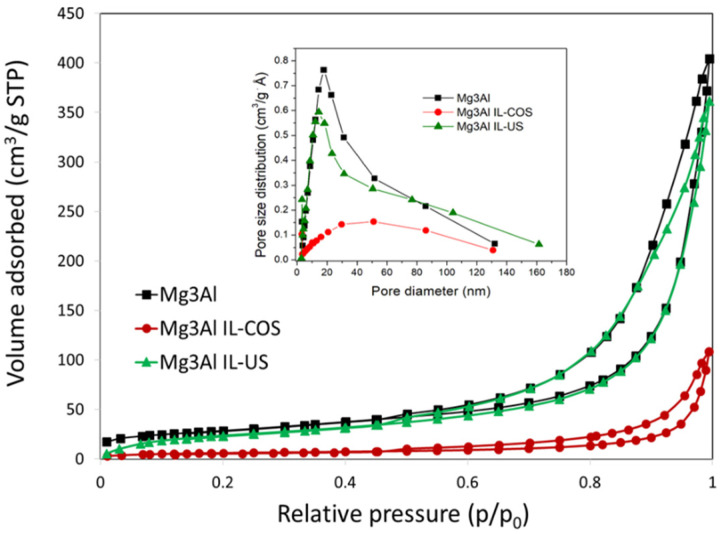
N_2_ adsorption-desorption isotherms. Inlet: pore size distribution.

**Figure 5 ijms-23-09107-f005:**
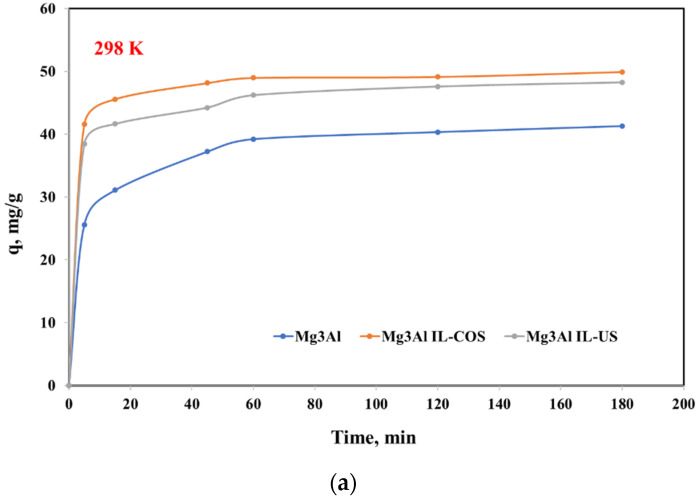
Effect of contact time on the adsorption capacity of studied LDHs in the adsorption process of Pd(II) ions from aqueous solutions at three different temperatures: (**a**) 298 K, (**b**) 313 K, (**c**) 328 K.

**Figure 6 ijms-23-09107-f006:**
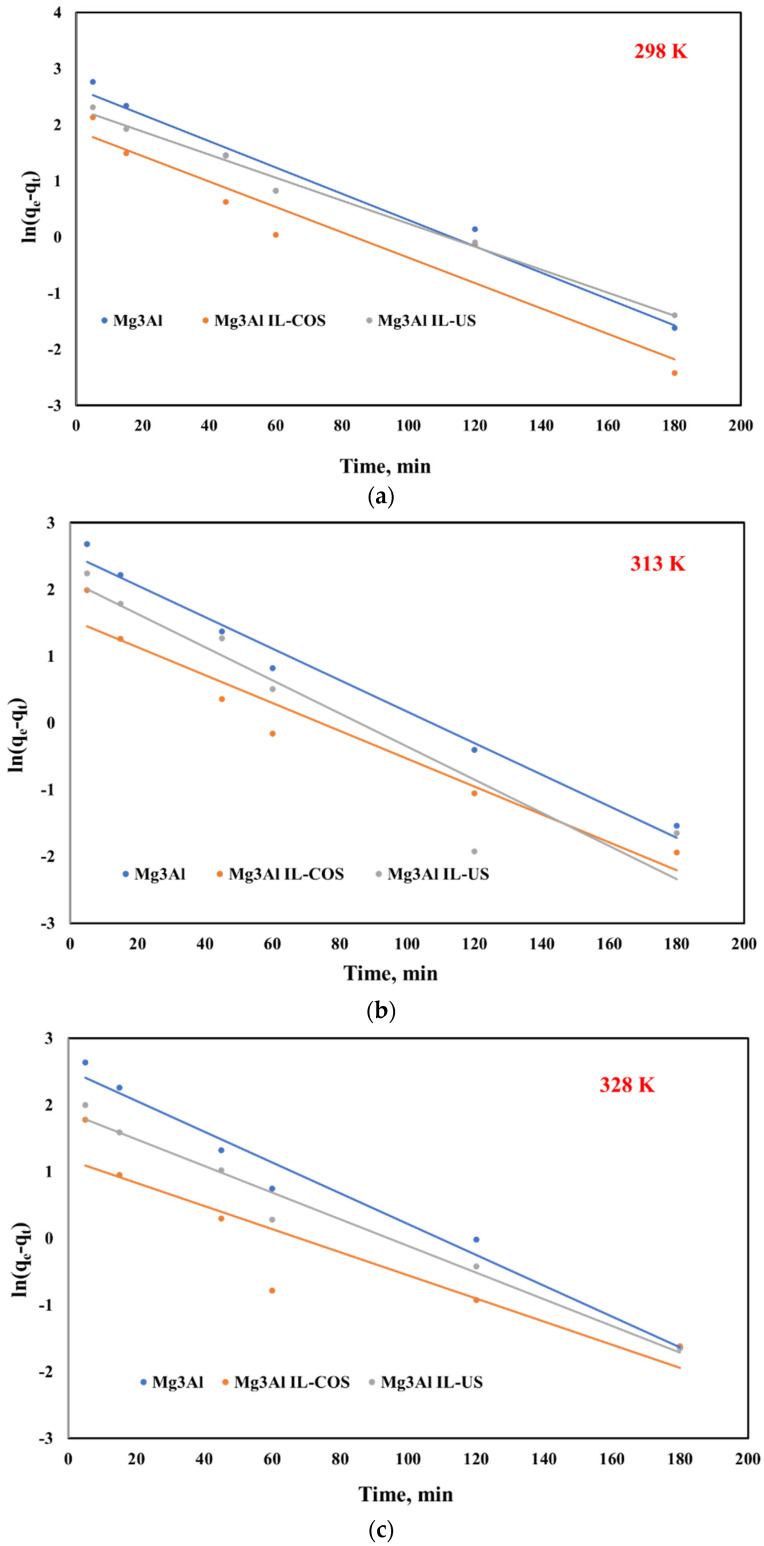
Pseudo-first order model for Pd(II) ions adsorption onto studied LDHs at three different temperatures: (**a**) 298 K, (**b**) 313 K, (**c**) 328 K.

**Figure 7 ijms-23-09107-f007:**
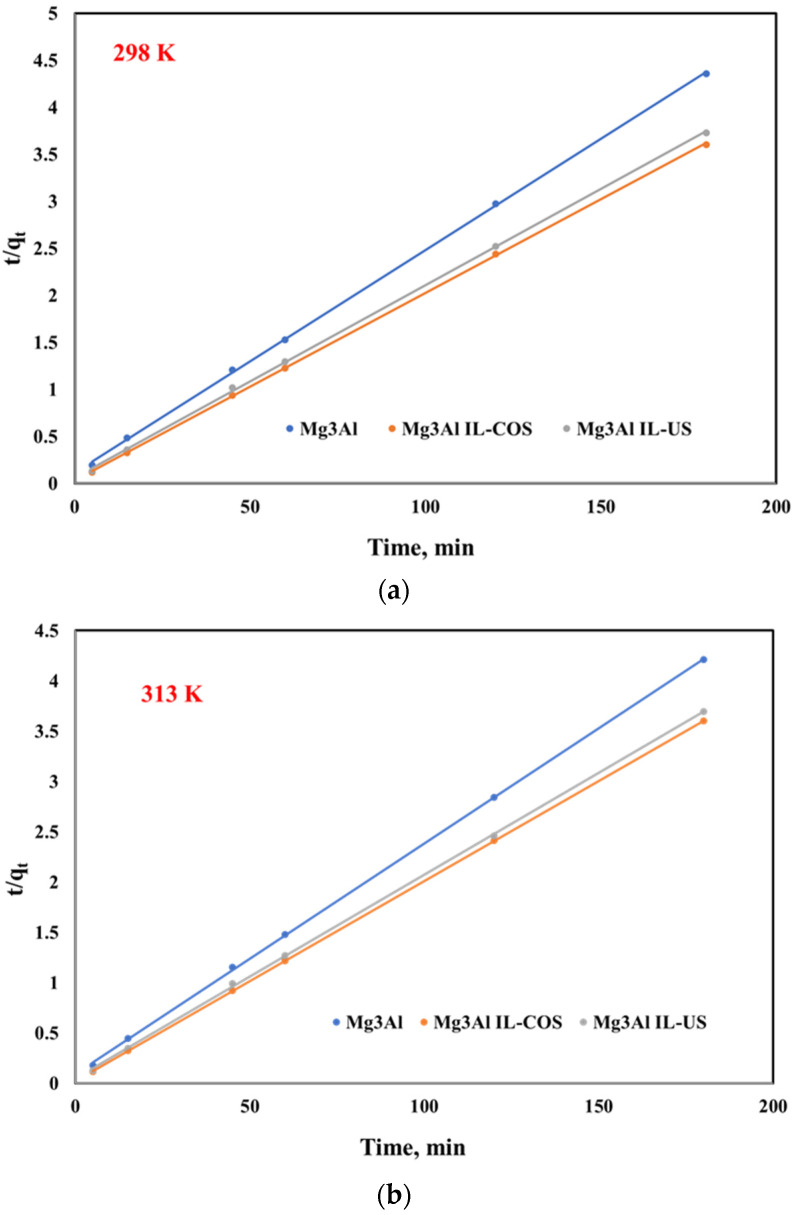
Pseudo-second order model for Pd(II) ions adsorption onto studied LDHs at three different temperatures: (**a**) 298 K, (**b**) 313 K, (**c**) 328 K.

**Figure 8 ijms-23-09107-f008:**
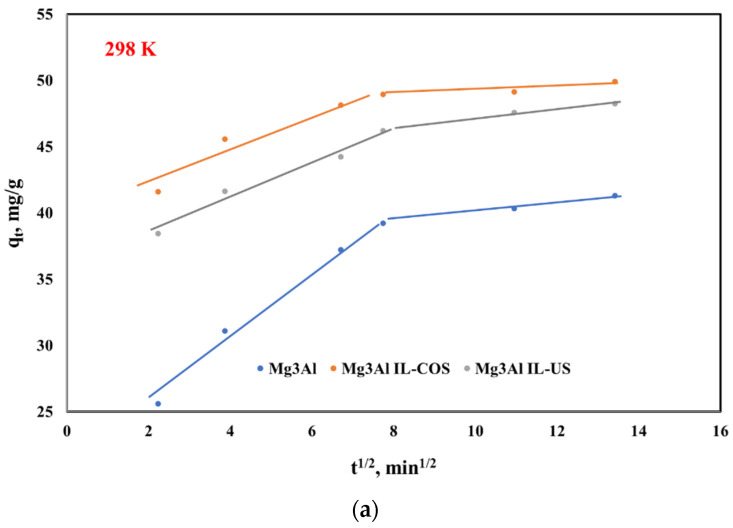
Intraparticle diffusion model for Pd(II) ions adsorption onto studied LDHs at three different temperatures: (**a**) 298 K, (**b**) 313 K, (**c**) 328 K.

**Figure 9 ijms-23-09107-f009:**
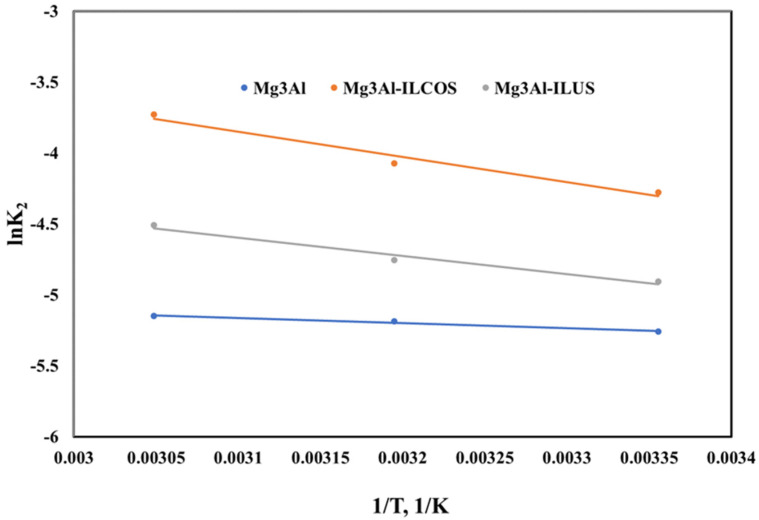
Arrhenius plot for Pd adsorption onto the studied LDHs.

**Figure 10 ijms-23-09107-f010:**
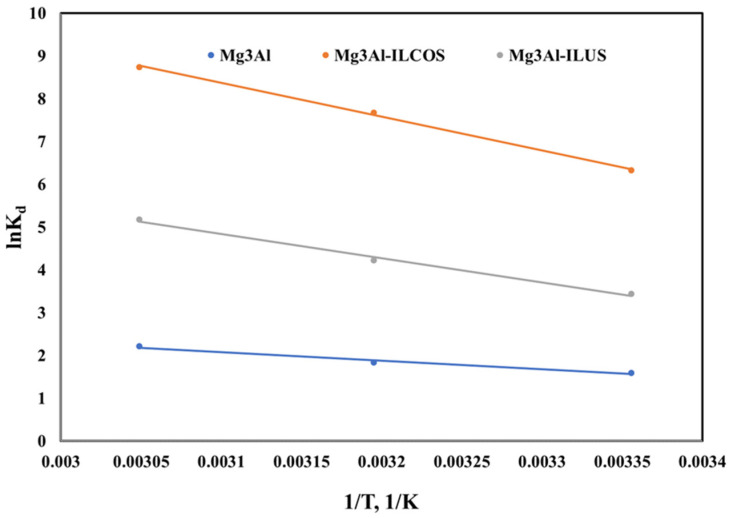
Effect of temperature on Pd adsorption onto the studied LDHs.

**Figure 11 ijms-23-09107-f011:**
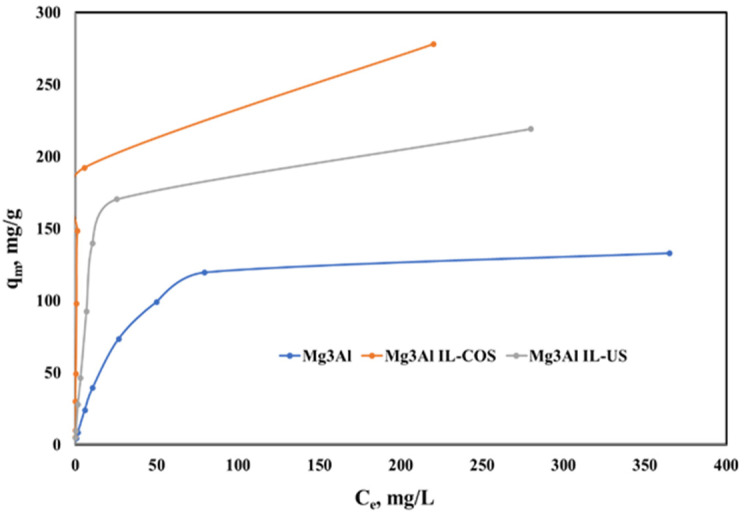
Equilibrium isotherms of Pd(II) ions adsorption onto studied LDHs.

**Figure 12 ijms-23-09107-f012:**
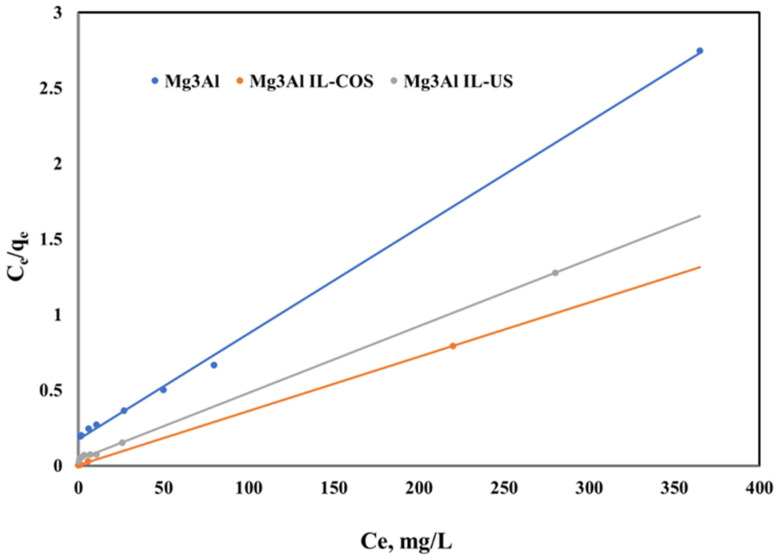
Langmuir isotherms of Pd(II) ions adsorption onto studied LDHs.

**Figure 13 ijms-23-09107-f013:**
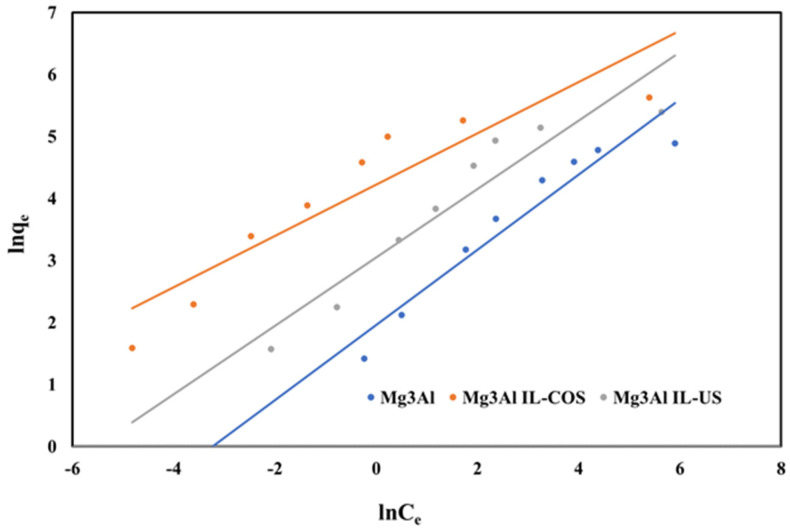
Freundlich isotherms of Pd(II) ions adsorption onto studied LDHs.

**Figure 14 ijms-23-09107-f014:**
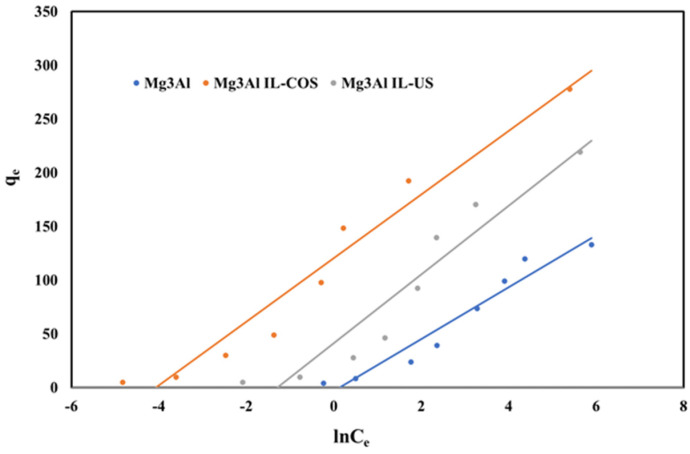
Temkin isotherms of Pd(II) ions adsorption onto studied LDHs.

**Figure 15 ijms-23-09107-f015:**
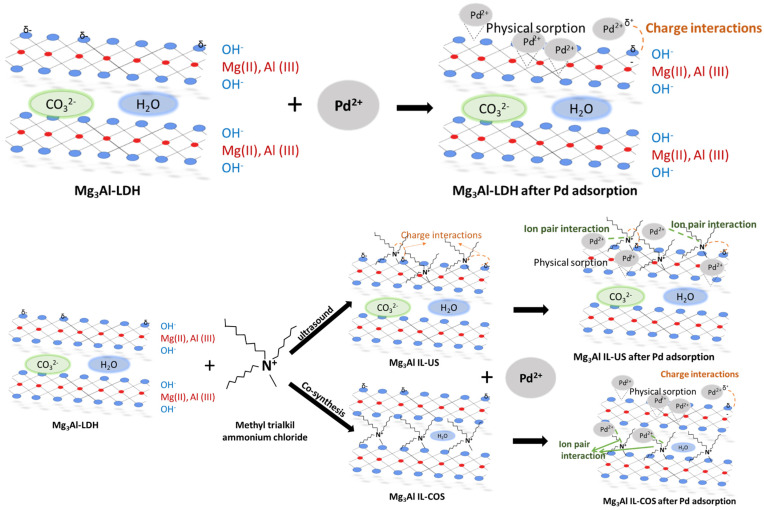
Adsorption mechanism of Pd ions onto the studied materials.

**Table 1 ijms-23-09107-t001:** Unit cell parameters and textural properties of the synthetized samples.

Method of Synthesis	Sample Acronym	a (Å)	c (Å)	d_(003)_ (Å)	D (nm)	S_BET_ (m^2^/g)	V_p_ (cm^3^/g)
MgAl-layered double hydroxide synthesized through co-precipitation Mg:Al = 3:1	Mg_3_Al	3.073	23.58	7.86	4.86	102	0.628
Mg_3_Al-LDH + IL functionalized by ultrasound method	Mg_3_Al IL-US	3.071	23.52	7.84	4.45	93.9	0.560
Mg_3_Al-LDH + IL functionalized by co-synthesis	Mg_3_Al IL-COS	3.072	24.39	8.13	3.38	22.2	0.168

**Table 2 ijms-23-09107-t002:** Kinetic models parameters for Pd(II) ions adsorption onto studied LDHs.

Adsorbent Material	Pseudo First Order				Pseudo Second Order			Intra Particle Diffusion		
	T = 298 K									
	q_e_exp. mg/g	q_e_ calc. mg/g	K_1,_ min^−1^	R^2^	q_e_ calc. mg/g	K_2_·10^−3^, min/(mg/g)	R^2^	K_int_, mg/g min^−1/2^	C	R^2^
**Mg_3_Al**	41.3	13.1	0.0288	0.9769	42.2	0.0052	0.9996	1.9826	23.8	0.9966
**Mg_3_Al IL-COS**	49.8	6.0	0.0218	0.9282	50.0	0.0139	0.999	0.9670	41.6	0.9943
**Mg_3_Al IL-US**	48.2	9.1	0.0198	0.9929	48.8	0.0074	0.9997	0.9976	38.0	0.9701
	**T = 313 K**									
	**q_e_exp.**	**q_e_ calc.**	**K_1_**	**R^2^**	**q_e_ calc.**	**K_2_**	**R^2^**	**K_int._**	**C**	**R^2^**
**Mg_3_Al**	42.7	12.5	0.0236	0.9837	43.7	0.0056	0.9998	1.2275	28.6	0.8383
**Mg_3_Al IL-COS**	49.9	4.7	0.0209	0.9384	50.3	0.017	1	0.5635	43.6	0.7448
**Mg_3_Al IL-US**	48.7	10.1	0.0248	0.8877	49.3	0.0086	0.9998	0.8110	39.3	0.8731
	**T = 328 K**									
	**q_e_exp.**	**q_e_ calc.**	**K_1_**	**R^2^**	**q_e_ calc.**	**K_2_**	**R^2^**	**K_int._**	**C**	**R^2^**
**Mg_3_Al**	44.7	12.4	0.0231	0.9761	45.5	0.0058	0.9998	1.1962	30.9	0.8406
**Mg_3_Al IL-COS**	50.0	3.3	0.0173	0.8254	50.3	0.0240	1	0.4379	45.1	0.7061
**Mg_3_Al IL-US**	49.7	6.6	0.0200	0.9756	50.0	0.0110	0.9999	0.6200	42.4	0.8752

**Table 3 ijms-23-09107-t003:** Thermodynamic parameters of Pd(II) adsorption onto the studied LDHs.

Adsorbent Material	E_a_, kJ/mol	Thermodynamic Parameters					
		ΔH°, kJ/mol^−1^	ΔS°, J/(mol K)	ΔG°, kJ/mol			R^2^
				298 K	313 K	328 K	
**Mg_3_Al**	2.97	16.56	68.2	−3.87	−4.9	−5.93	0.9784
**Mg_3_Al IL−COS**	14.72	65.38	272.2	−15.6	−19.7	−23.8	0.9983
**Mg_3_Al IL−US**	10.68	46.98	185.9	−8.41	−11.2	−13.9	0.9926

**Table 4 ijms-23-09107-t004:** Equilibrium sorption isotherm parameters for Pd(II) ions adsorption onto studied LDHs.

		Mg_3_Al	Mg_3_Al IL-COS	Mg_3_Al IL-US
Equilibrium isotherm	q_m,_ exp, mg/g	132.9	277.8	219.1
Langmuir	q_m,_ calc, mg/g	142.9	277.7	227.3
	K_L_, L/mg	0.0399	0.8782	0.1073
	R^2^	0.9987	0.9999	0.9994
Freundlich	1/n	0.6063	0.4135	0.5508
	K_F_, mg/g	7.11	68.3	21.2
	R^2^	0.9214	0.8515	0.8926
Temkin	b, J/mol	102.8	83.7	77.6
	K_T_, L/g	0.872	58.8	3.63
	R^2^	0.9432	0.9536	0.91

**Table 5 ijms-23-09107-t005:** Comparison of some of the selected diverse adsorbents reported in recent literature.

Adsorbent	q_m_, mg/g	References
2-Mercaptobenzothiazole functionalized Amberlite XAD-1180 resin	50.0	[46]
Silica-based adsorbent functionalized with macrocyclic ligand	83.0	[47]
Aliquat-336 (ionic liquid) impregnated onto chitosan	187.61	[48]
Tetraoctylammonium bromide impregnated onto graphene oxide	92.67	[49]
MgSiO_3_functionalizedwith DL-cysteine	9.23	[52]
Zn_3_Al	64.4	[23]
Zn_3_Al-IL	100	
Zn_3_Al-ILUS	92.4	
Mg_3_Al	142.9	Present paper
Mg_3_Al IL-COS	277.8	
Mg_3_Al IL-US	227.3	

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
