# Peer review of "IL-Functionalized Mg3Al-LDH as New Efficient Adsorbent for Pd Recovery from Aqueous Solutions"

_ijms, 2022, doi:10.3390/ijms23169107_

Round 1

Reviewer 1 Report

This paper describes the IL-functionalized Mg3Al-LDH as an efficient adsorbent for Pd recovery from aqueous solutions.

Organization of the manuscript is very good. The topic has been practically outlined in this document, but there are also errors and English language mistakes. The results discussion should be improved and some experiments should be conducted.

The main mistakes and the suggestions have been given below sequentially.

1) In Abstract section more numerical values need to be added.

2) In the Introduction section the purpose of the work should be clearly explained.

3) The characterization of the adsorbents should be improved. The surface morphology of the adsorbents also should be presented in the manuscript.

4) Please do also column adsorption experiments.

5) In the manuscript the discussion is not deep enough. Results discussion should be extended.

6) The adsorption mechanism study should be added in the Results and Discussion section.

7) The reusability evaluation and stability of the adsorbent tests should be done and discussed in the manuscript.

8) The comparison of adsorbent with others materials is poor and it should be highly improved.

9) In Conclusion section the future perspectives, impact of the work and obtained results on the audience should be presented.

10) The IL chemical nomenclature should be checked and improved.

Author Response

This paper describes the IL-functionalized Mg3Al-LDH as an efficient adsorbent for Pd recovery from aqueous solutions.

Organization of the manuscript is very good. The topic has been practically outlined in this document, but there are also errors and English language mistakes. The results discussion should be improved and some experiments should be conducted.

The main mistakes and the suggestions have been given below sequentially.

1) In Abstract section more numerical values need to be added.

In the abstract section more numerical values has been added.

2) In the Introduction section the purpose of the work should be clearly explained.

The purpose of the work was clearly explained in the introduction section.

3) The characterization of the adsorbents should be improved. The surface morphology of the adsorbents also should be presented in the manuscript.

The surface morphology of the studied sample has been published before, when these materials were used for diclofenac adsorption from aqueous solutions (DOI: 10.3390/molecules26237384). In the paper we discussed about the morphological analysis with reference to the already published work. We consider that the X-ray diffraction (XRD), N2 adsorption-desorption analysis, Fourier-transform infrared (FTIR) spectroscopy and thermal analysis represents a great number of procedures to elucidate the material structure and behavior.

4) Please do also column adsorption experiments.

The column adsorption experiments represent a complex and laborious study which could be the subject of a new paper. The aim of this work was to synthesize and characterize these new adsorbent materials and to determine the influence of the synthesis method on the adsorption capacity developed by the obtained materials in the recover process of Pd ions from aqueous solutions.

5) In the manuscript the discussion is not deep enough. Results discussion should be extended.

The results and discussion were extended.

6) The adsorption mechanism study should be added in the Results and Discussion section.

The adsorption mechanism study was introduced and detailed discussed in the Results and Discussion,  section 2.5.

7) The reusability evaluation and stability of the adsorbent tests should be done and discussed in the manuscript.

Our purpose is to reclaim and further use of the exhausted adsorbent as photocatalyst (due to well-known photocatalytic properties of palladium ions) in the degradation process of organic compounds from waters, considering that we already have results in this purpose (https://doi.org/10.1016/j.seppur.2020.117104). This part will be the subject of a future work, and these was mentioned in the conclusion part as a future perspective.

8) The comparison of adsorbent with others materials is poor and it should be highly improved.

The comparison of adsorbent with other materials was highly improved in the Results and Discussion,  section 2.6.

9) In Conclusion section the future perspectives, impact of the work and obtained results on the audience should be presented.

In the conclusion section the future perspective, impact of the work and obtained results on the audience were presented.

10) The IL chemical nomenclature should be checked and improved.

The IL chemical nomenclature was checked, and it is correct.

Reviewer 2 Report

The manuscript reports the adsorption of Pd by means of IL-Functionalized Mg3Al-LDH.

Physico-chemical characterization of the Mg3Al-LDH materials has been well performed and presented. Kinetic, equilibrium and thermodynamic studies have been correctly carried out and the data have been well discussed. In addition, the adsorbent material used in this manuscript has been compared with similar ones reported in the literature in the removal process of Pd ions from aqueous solutions.

In my opinion, the manuscript "IL-Functionalized Mg3Al-LDH as New Efficient Adsorbent for Pd Recovery from Aqueous Solutions" could be accepted for publication after minor revision, with the following concerns addressed:

-          In the Abstract (line 14), the Authors report that the “IL (methyl trialkyl ammonium chloride)” is used to functionalize the Mg3Al-LDH. However, Authors should clarify the meaning of acronym IL the first time it appears in the text.

-          The authors should correct the typo in line 127 pg. 4.

-        In the session “2. Results and Discussions” the Authors report the characterization of Mg3Al IL-US and Mg3Al IL-COS adsorbent materials. However, the meaning of their names is described in the following paragraph “3. Materials and Methods” and it makes difficult the text reading.

          -    There is a typo in line 111 pg. 19.

-                 -    To have very efficient materials for adsorption processes, absorbents             should also be reused in subsequent adsorption cycles. The authors               believe that the studied Mg3Al–LDH materials can carry out                             desorption and subsequent re-adsorption processes of Pd(II) for a very           efficient recyclability of the adsorbent material? I suggest adding some           comments about this topic.

Author Response

The manuscript reports the adsorption of Pd by means of IL-Functionalized Mg3Al-LDH.

Physico-chemical characterization of the Mg3Al-LDH materials has been well performed and presented. Kinetic, equilibrium and thermodynamic studies have been correctly carried out and the data have been well discussed. In addition, the adsorbent material used in this manuscript has been compared with similar ones reported in the literature in the removal process of Pd ions from aqueous solutions.

In my opinion, the manuscript "IL-Functionalized Mg3Al-LDH as New Efficient Adsorbent for Pd Recovery from Aqueous Solutions" could be accepted for publication after minor revision, with the following concerns addressed:

-          In the Abstract (line 14), the Authors report that the “IL (methyl trialkyl ammonium chloride)” is used to functionalize the Mg3Al-LDH. However, Authors should clarify the meaning of acronym IL the first time it appears in the text.

-  the meaning of acronym IL was introduced

-          The authors should correct the typo in line 127 pg. 4.

-  the typo on page 4 was corrected.

-        In the session “2. Results and Discussions” the Authors report the characterization of Mg3Al IL-US and Mg3Al IL-COS adsorbent materials. However, the meaning of their names is described in the following paragraph “3. Materials and Methods” and it makes difficult the text reading.

- the meaning of the sample names was introduced in Table 1.

          -    There is a typo in line 111 pg. 19.

- the typo was corrected

-       To have very efficient materials for adsorption processes, absorbents             should also be reused in subsequent adsorption cycles. The authors               believe that the studied Mg3Al–LDH materials can carry out                             desorption and subsequent re-adsorption processes of Pd(II) for a very           efficient recyclability of the adsorbent material? I suggest adding some           comments about this topic.

- Yes this is true, but our purpose is to reclaim and further use of the exhausted adsorbent as photocatalyst (due to well known photocatalytic properties of palladium ions) in the degradation process of organic compounds from waters, taking into account that we already have results in this purpose (https://doi.org/10.1016/j.seppur.2020.117104).

Reviewer 3 Report

Authors functionalized Mg3Al layered double hydroxide with methyl trialkyl ammonium chloride for enhancing adsorption capacity of palladium from aqueous solution. More professional analysis is required for their IR spectra. For example, which functional group and what kind of vibrational mode with any reliable reference literature. Comments and questions are the following.

1. Figure 1: Although XRD data are measured, any further analysis like peak assignments with reference literature wasn’t done. 

2. Figure 1: What is the reference JCPDS number of hydrotalcite?

3. Figure 1: They need to mention 2theta values of each sample. 

5. Figure 3: Authors assigned the peaks at 2920 and 2852 cm-1 as CH2 and CH3, respectively. Do they have any reference literature? In addition to the compound group, they need to mention which vibrational mode (stretching, vending, rocking, or deformation) exactly. Are those related to symmetric and asymmetric stretching vibrations?

6. Page 5, Line 159-160: Authors mentioned quaternary ammonium group is observed at 1462 and 1377 cm-1. They need to mention exactly which vibrational modes are related to the peaks. 

7. Figure 3: Authors seem to follow the assignments in Ref. [33], but Ref. [33] analyzed IR spectra without any reliable reference literature. Authors need to cite any other good reference literature for their analysis on the spectra in Figure 3.

Author Response

Authors functionalized Mg3Al layered double hydroxide with methyl trialkyl ammonium chloride for enhancing adsorption capacity of palladium from aqueous solution. More professional analysis is required for their IR spectra. For example, which functional group and what kind of vibrational mode with any reliable reference literature. Comments and questions are the following.

  1. Figure 1: Although XRD data are measured, any further analysis like peak assignments with reference literature wasn’t done. 

A peak assignments analysis with reference literature was done.

  1. Figure 1: What is the reference JCPDS number of hydrotalcite?

The number of the JCPDS of hydrotalcite was mentioned.

  1. Figure 1: They need to mention 2theta values of each sample. 

The 2theta values of each samples were mentioned.

  1. Figure 3: Authors assigned the peaks at 2920 and 2852 cm-1as CH2and CH3, respectively. Do they have any reference literature? In addition to the compound group, they need to mention which vibrational mode (stretching, vending, rocking, or deformation) exactly. Are those related to symmetric and asymmetric stretching vibrations?

The entire paragraph was changed according with the reviewer observations.  

  1. Page 5, Line 159-160: Authors mentioned quaternary ammonium group is observed at 1462 and 1377 cm-1. They need to mention exactly which vibrational modes are related to the peaks. 

The vibrational modes related to the peaks were mentioned.

  1. Figure 3: Authors seem to follow the assignments in Ref. [33], but Ref. [33] analyzed IR spectra without any reliable reference literature. Authors need to cite any other good reference literature for their analysis on the spectra in Figure 3.

The ref [33], now [36], was changed accordingly to the reviewer suggestion.

Round 2

Reviewer 1 Report

The manuscript has met improvements. However minor correction should be do.

1) The surface morphology of the studied sample should be shortly discussed and previous work (DOI: 10.3390/molecules26237384) should be cited.

2) The IL name should be: Methyltrialkylammonium chloride.

3) In conclusion section the information about the column adsorption test as a future experiment should be added.

4) English language and spell should be checked.

Author Response

1) The surface morphology of the studied sample should be shortly discussed and previous work (DOI: 10.3390/molecules26237384) should be cited.

The surface morphology of the studied sample was shortly discussed and the previous work was cited after the first review in the page 4 of the manuscript.

2) The IL name should be: Methyltrialkylammonium chloride.

The IL name was corrected in the entire manuscript.

3) In conclusion section the information about the column adsorption test as a future experiment should be added.

Information about the column adsorption test as a future experiment were added in the conclusion section.

4) English language and spell should be checked.

English language and spell was checked and corrected.